# Impact of Diverse Parameters on the Physicochemical Characteristics of Green-Synthesized Zinc Oxide–Copper Oxide Nanocomposites Derived from an Aqueous Extract of *Garcinia mangostana* L. Leaf

**DOI:** 10.3390/ma16155421

**Published:** 2023-08-02

**Authors:** Yu Bin Chan, Mohammod Aminuzzaman, Lai-Hock Tey, Yip Foo Win, Akira Watanabe, Sinouvassane Djearamame, Md. Akhtaruzzaman

**Affiliations:** 1Department of Chemical Science, Faculty of Science, Universiti Tunku Abdul Rahman (UTAR), Kampar Campus, Jalan Universiti, Bandar Barat, Kampar 31900, Malaysia; yubinchan1221@gmail.com (Y.B.C.); yipfw@utar.edu.my (Y.F.W.); 2Centre for Photonics and Advanced Materials Research (CPAMR), Universiti Tunku Abdul Rahman (UTAR), Sungai Long Campus, Jalan Sungai Long, Bandar Sungai Long, Kajang 43000, Malaysia; 3Institute of Multidisciplinary Research for Advanced Materials (IMRAM), Tohoku University, Sendai 980-8577, Japan; akira.watanabe.c6@tohoku.ac.jp; 4Department of Biomedical Science, Faculty of Science, Universiti Tunku Abdul Rahman (UTAR), Kampar Campus, Jalan Universiti, Bandar Barat, Kampar 31900, Malaysia; sinouvassane@utar.edu.my; 5Solar Energy Research Institute (SERI), Universiti Kebangsanan Malaysia (UKM), Bangi 43600, Malaysia; akhtar@ukm.edu.my

**Keywords:** *Garcinia mangostana* L., green product, green synthesis, nanocomposites, zinc oxide-copper oxide

## Abstract

Compared to conventional metal oxide nanoparticles, metal oxide nanocomposites have demonstrated significantly enhanced efficiency in various applications. In this study, we aimed to synthesize zinc oxide–copper oxide nanocomposites (ZnO-CuO NCs) using a green synthesis approach. The synthesis involved mixing 4 g of Zn(NO_3_)_2_·6H_2_O with different concentrations of mangosteen (*G. mangostana*) leaf extract (0.02, 0.03, 0.04 and 0.05 g/mL) and 2 or 4 g of Cu(NO_3_)_2_·3H_2_O, followed by calcination at temperatures of 300, 400 and 500 °C. The synthesized ZnO-CuO NCs were characterized using various techniques, including a UV-Visible spectrometer (UV-Vis), photoluminescence (PL) spectroscopy, Fourier Transform Infrared (FTIR) spectroscopy, X-ray powder diffraction (XRD) analysis and Field Emission Scanning Electron Microscope (FE-SEM) with an Energy Dispersive X-ray (EDX) analyzer. Based on the results of this study, the optical, structural and morphological properties of ZnO-CuO NCs were found to be influenced by the concentration of the mangosteen leaf extract, the calcination temperature and the amount of Cu(NO_3_)_2_·3H_2_O used. Among the tested conditions, ZnO-CuO NCs derived from 0.05 g/mL of mangosteen leaf extract, 4 g of Zn(NO_3_)_2_·6H_2_O and 2 g of Cu(NO_3_)_2_·3H_2_O, calcinated at 500 °C exhibited the following characteristics: the lowest energy bandgap (2.57 eV), well-defined Zn-O and Cu-O bands, the smallest particle size of 39.10 nm with highest surface area-to-volume ratio and crystalline size of 18.17 nm. In conclusion, we successfully synthesized ZnO-CuO NCs using a green synthesis approach with mangosteen leaf extract. The properties of the nanocomposites were significantly influenced by the concentration of the plant extract, the calcination temperature and the amount of precursor used. These findings provide valuable insights for researchers seeking innovative methods for the production and utilization of nanocomposite materials.

## 1. Introduction

Compared to individual semiconductor metal oxide nanoparticles (NPs), such as zinc oxide (ZnO), copper oxide (CuO), nickel oxide (NiO), etc., the mixing of these NPs has gained significant attention due to their excellent application in sensor, electrical and electronic products. Mixing semiconductor metal oxides allows for control over their structural, morphological and surface properties, making them important in various practical applications [1]. Among the *p*-*n* type mixed semiconductors, ZnO-CuO nanocomposites (NCs) garnered considerable interest from researchers. Copper is preferred to combine with ZnO due to its ability to easily overlap *d*-electrons with a valence bond of ZnO [2]. This results in enhanced surface area, smaller particle size and the formation of ZnO-CuO heterojunctions, which strengthen the optical and electronic properties [2,3]. Consequently, ZnO-CuO NCs find application in environmental remediation, photo-catalysis, fuel cell, solar cell, antibacterial, UV protection and optoelectronics devices [1,4,5,6]. For example, the effectiveness in degrading methylene blue was higher by using ZnO-CuO NCs (98%) compared to ZnO (81%) [5].

Green synthesis of nanomaterial offers a simpler, more cost-effective, eco-friendly alternative with lower energy consumption compared to conventional methods [7,8,9,10,11,12]. Generally, various parts of plants, including flowers, leaves, stems, roots and seeds, are utilized in the green synthesizing of nanomaterials [13,14,15]. During the green synthesis process, phytochemicals present in plants, such as phenols, aldehydes, ketones, carboxylic acids, nitrogenous compounds, flavonoids, alkaloids, terpenoids, tannins and pigments, accumulate and later interact with metals to cap, stabilize and reduce to NPs [9,16,17]. However, achieving the desired morphology and shape remains a challenge in the green synthesis of NPs and NCs. As a result, extensive research has been conducted to optimize the synthesis conditions, including plant extract concentration, temperature and precursor concentration, to synthesize NPs and NCs with desired structural, morphological and optical properties [18,19].

While aqueous extract from *Aloe barbadansis* leaf [3], *Calotropis gigantea* leaf [4], *Theobroma cacao* seed bark [6], *Dovyalis caffra* leaf [20], *Verbascum sinaiticum* Benth [21], *Sambucus nigra* L. shoot [22], *Alchornea cordifolio* leaf [23] and *Calotropis gigamtae* leaf [24] has been utilized for synthesizing ZnO-CuO NCs. The use of *Garcinia mangostana* L., commonly known as mangosteen, in synthesizing ZnO-CuO NCs has not been explored. Mangosteen is a seasonal fruit in the *Clusiacae* family and is commonly found in tropical countries [25,26,27,28,29]. It contains numerous phytochemicals, such as xanthones, flavonoids and terpene [30,31,32,33], which have the potential to form stable colloidal nanomaterials.

In this study, we synthesized ZnO-CuO NCs using a mangosteen leaf aqueous extract in a green, fast and simple manner. The mangosteen leaf aqueous extract-mediated ZnO-CuO NCs were optimized by varying the concentration of the mangosteen leaf aqueous extract (0.02, 0.03, 0.04 and 0.05 g/mL), calcination temperatures (300, 400 and 500 °C) and the amount of Cu(NO_3_)_2_·3H_2_O (2.0 and 4.0 g). In this paper, we investigated the effects of these parameters (plant concentration, calcination temperature and precursor weight) on the optical, structural and morphological properties of the mangosteen leaf aqueous extract-mediated ZnO-CuO NCs.

## 2. Materials and Methods

### 2.1. Materials

The mangosteen leaves were collected from a neighborhood in Kampar, Malaysia. Zinc nitrate hexahydrate, Zn(NO_3_)_2_·6H_2_O, was purchased from HiMedia Laboratories Pvt. Ltd. (Nashik, India), and copper nitrate trihydrate, Cu(NO_3_)_2_·3H_2_O was purchased from HmbG (Hamburg, Germany). Both chemicals were used without further purification. All glassware was washed with deionized water and dried in an oven before use.

### 2.2. Characterization

The selection of optimized parameters in green synthesizing ZnO-CuO NCs was based on their structural, morphological and optical properties. The absorption spectra were recorded by a UV-Visible (UV-Vis) spectrophotometer (Thermo Scientific GENESYS 10S, Waltham, MA, USA). The recombination of electron-hole pairs (e^−^/h^+^) of the synthesized samples was investigated using photo luminance (PL) spectroscopy (Perkin Elmer LS 55 Fluorescence Spectrometer, Waltham, MA, USA) with an excitation wavelength of 350 nm in the range of 350 to 600 nm. The Fourier Transform Infrared (FTIR) spectroscopy study was carried out at room temperature in the range of 4000 to 400 cm^−1^ with a resolution of 4 cm^−1^ by using KBr pellets in a Perkin Elmer RX1 spectrophotometer. X-ray powder diffraction (XRD) patterns were taken in the reflection mode with Cu Kα (λ = 1.5406 Å) radiation in the 2*θ* range of 10° to 80° by using a Shimadzu XRD 6000 X-ray diffractometer with continuous scanning which was operated at 40 kV/30 mA and 0.02 min^−1^. The morphological, microstructural and elemental compositional of all synthesized samples was determined using a Field Emission Scanning Electron Microscope (FE-SEM) (JEOL JSM-6710F, Tokyo, Japan) with Energy Dispersive X-ray (EDX) analyzer (X-max, 150 Oxford Instruments, Abingdon-*on*-Thames, UK).

### 2.3. Preparation of Mangosteen Leaf Aqueous Extract

The freshly plucked mangosteen leaves were washed with tap water to remove dust and dried in an oven at 50 °C for 48 h and further dried in a vacuum oven at 60 °C for 8 h. Then, the leaves were ground into a fine powder by using a grinder. Then, 5 g of leaf powder was added to 100 mL of deionized water and heated with stirring at 70–80 °C for 20 min to obtain 0.05 g/mL of leaf aqueous extract. Upon cooling, the leaf aqueous extract was vacuum filtrated, and a reddish-brown filtrate was collected and immediately used for ZnO-CuO NCs synthesis.

### 2.4. Synthesis of ZnO-CuO NCs

With minor modification from Chan et al. [34], the synthesis of ZnO-CuO NCs using mangosteen leaf aqueous extract was performed. The reaction parameters, which included mangosteen leaf aqueous extract concentration, calcination temperature and weight of Cu(NO_3_)_2_·3H_2_O added, were optimized.

#### 2.4.1. Leaf Aqueous Extract Optimization

The 50 mL of mangosteen leaf aqueous extract (0.02, 0.03, 0.04 and 0.05 g/mL) was mixed separately with 4.0 g of Zn(NO_3_)_2_·6H_2_O and 2 g of Cu(NO_3_)_2_·3H_2_O. Immediately, a greenish-brown solution formed. The solution was heated at 70–80 °C with constant stirring until the formation of a brown paste. The paste was then cooled to room temperature and calcinated at 500 °C for 2 h using the Muffle furnace to obtain a fine black-blue ZnO-CuO powder.

#### 2.4.2. Calcination Temperature Optimization

After the selection of the optimized mangosteen leaf aqueous extract concentration at 0.05 g/mL, the synthesis of ZnO-CuO NCs was repeated using 4 g of Zn(NO_3_)_2_·6H_2_O and 2 g of Cu(NO_3_)_2_·3H_2_O. The cooled brown paste was calcinated at 300, 400 and 500 °C for 2 h to have more energy savings during the ZnO-CuO NCs synthesis.

#### 2.4.3. Precursor Optimization

After the selection of the optimized mangosteen leaf aqueous extract concentration at 0.05 g/mL and calcination temperature at 500 °C, the synthesis steps were repeated using 4 g of Zn(NO_3_)_2_·6H_2_O with different weights of Cu(NO_3_)_2_·3H_2_O (2 and 4 g). Until the formation of brown paste. It was then calcinated at 500 °C for 2 h.

## 3. Results

### 3.1. UV-Vis Spectroscopy Analysis

Figure 1 shows the UV-Vis spectra of the mangosteen leaf aqueous extract, Cu(NO_3_)_2_·3H_2_O, Zn(NO_3_)_2_·6H_2_O and mangosteen leaf aqueous extract-mediated ZnO-CuO NCs with their energy bandgap. The absorption peak position had no significant changes in ZnO-CuO NCs synthesized at different controlled parameters. The mangosteen leaf aqueous extract absorption peak was located at 479 cm^−1^, while for Cu(NO_3_)_2_·3H_2_O and Zn(NO_3_)_2_·6H_2_O, it was located at 295 and 305 cm^−1^, respectively. On the other hand, the ZnO-CuO NCs absorption peak was located at 369–375 cm^−1^.

The energy bandgap of the mangosteen leaf aqueous extract-mediated ZnO-CuO NCs at different synthesizing conditions is tabulated in Table 1. The energy band gap of the ZnO-CuO NPs was expressed in eV and calculated using a Tauc-plot approach using Equation (1).
(1)$$\alpha hv=A{\left(hv-E_{g}\right)}^{n}$$
where *h* is Plank’s constant (6.626 × 10^−34^ Js), *n* is the exponential factor for electronic transition (*n* = ½ for the indirect band, *n* = 2 for the direct band) and α is the absorption coefficient. The energy bandgap showed no significant difference when using 2 g (2.57 eV) and 4 g (2.56 eV) of Cu(NO_3_)_2_·3H_2_O in synthesizing ZnO-CuO NCs. In contrast, a plant aqueous extract concentration-dependent and calcination temperature-dependent shifts were observed as the energy bandgap decreased from 3.31 eV to 2.57 eV and higher leaf aqueous extract concentrations and calcination temperatures were applied. 

### 3.2. FTIR Spectroscopy Analysis

The FTIR spectra interpretation of mangosteen leaf aqueous extract-mediated ZnO-CuO NCs at different controlled parameters is shown in Table 2, and their FTIR spectra are shown in Figure 2. The 3401–3436 cm^−1^ and 1629–1636 cm^−1^ bands corresponded to *v*(O-H) and *v*(C=O) or *v*(C=C). Moreover, 1384 cm^−1^ and 1099–1114 cm^−1^ bands were assigned to *v*(C-C aromatic) and *v*(C-O). The bond vibration of CuO and ZnO was indicated by the bands at 649–674 cm^−1^ and 447–524 cm^−1^, respectively.

The *v*(C-C aromatic) and *v*(C=O) or *v*(C=C) intensities increased when higher concentrations of mangosteen leaf aqueous extract were used. On the other hand, v(C-C aromatic) and *v*(C=O) or *v*(C=C) intensities decreased, while *v*(Cu-O) intensity increased at elevated calcination temperatures. Additionally, the bands, which included *v*(C-C aromatic), *v*(C=O) or *v*(C=C), *v*(C-O) and *v*(Cu-O) intensities improved when more Cu(NO_3_)_2_·3H_2_O was added.

### 3.3. PL Spectroscopy Analysis

The potential recombination of the photo-generated electron-hole (e^−^/h^+^) pairs of the mangosteen leaf aqueous extract-mediated ZnO-CuO NCs and the occurrence of their electronic transfer in NCs were determined by using PL spectroscopy (Figure 3). Overall, the ZnO-CuO NCs emission peaked in the violet region (390–405 nm). From Figure 3, it can be observed that the PL intensity was more affected by calcination temperature as high temperature-calcinated ZnO-CuO NCs had lower charge carrier separation compared to lower temperature samples.

### 3.4. XRD Spectroscopy Analysis

The mangosteen leaf aqueous extract-mediated ZnO-CuO NCs with the reference card number ICDD 01-081-9217 were in a hexagonal-wurtzite phase, *a* = 3.2459 Å and *c* = 5.1975 Å, with space group *P*63*mc*. All peaks were very sharp and intense, indicating the samples were of a crystalline nature. ZnO-CuO NCs had diffraction peaks at 2*θ* values of 31.74, 34.41, 36.22, 47.57, 56.58, 62.90 and 69.03°, matched with the ZnO phase, indexed as (1 0 0), (0 0 2), (1 0 1), (1 0 2), (1 1 0), (1 0 3) and (2 0 1), respectively. Meanwhile, those 2*θ* values of 32.60, 35.58, 38.80, 48.59, 61.57, 66.30 and 68.01^o^ matched with CuO phase were indexed as (−1 1 0), (0 0 2), (1 1 1), (−2 0 2), (−1 1 3), (0 2 2) and (−2 2 0), respectively. On the other hand, the CuO-indexed peaks’ intensity magnitude was highest when using 4 g of Cu(NO_3_)_2_·3H_2_O, especially (1 1 1). The ZnO-CuO NCs spectra are shown in Figure 4.

The crystallinity of green-synthesized ZnO-CuO NCs was significantly affected by the mangosteen leaf aqueous concentration compared to calcination temperature and added Cu(NO_3_)_2_·3H_2_O weight (Table 3). As shown in Equation (2), Debye–Scherrer's formula was used to calculate the crystalline size of ZnO-CuO NCs [35].
(2)$$D=\frac{0.94\lambda }{\beta cos\theta }$$
where *D* is the crystalline size of NPs, *λ* is the X-ray wavelength, *β* is the full-width half-maximum (FWHM) of the peak and *θ* is the Bragg angle. In general, the crystalline size of the ZnO-CuO NCs was in the range of 18.17 to 28.51 nm. The decrement in crystalline size of ZnO-CuO NCs at elevated mangosteen leaf aqueous extract concentrations (from 28.51 nm to 18.17 nm) and calcination temperature (from 22.25 nm to 18.17 nm) was obtained. In contrast, a slight increment in the crystalline size of ZnO-CuO NCs, from 18.17 nm to 22.29 nm, when the weight of the added Cu(NO_3_)_2_·3H_2_O increased from 2 g to 4 g.

The ZnO-CuO NCs dislocation density was estimated using Williamson and Smallman’s formula [35] in Equation (3).
(3)$$\delta =\frac{1}{D^{2}}$$
where *δ* is the dislocation density of NPs and *D* is the NPs’ crystalline size. The ZnO-CuO NCs’ dislocation density was in the range of 12.31 × 10^14^ to 30.30 × 10^14^ cm^−1^. An increment in dislocation density was obtained when higher mangosteen leaf aqueous extract concentrations (from 12.31 × 10^14^ cm^−1^ to 30.30 × 10^14^ cm^−1^) and calcination temperatures (from 20.19 × 10^14^ cm^−1^ to 30.30 × 10^14^ cm^−1^) were applied. However, their dislocation density decreased from 30.30 × 10^14^ cm^−1^ to 20.13 × 10^14^ when more Cu(NO_3_)_2_·3H_2_O was added during the green synthesis of ZnO-CuO NCs.

Equation (4) was used to calculate the micro strain of the ZnO-CuO NCs [35].
(4)$$\varepsilon =\frac{\beta cos\theta }{4}$$
where *ε* is the micro strain of NPs, *β* is the FWMH of the peak and *θ* is the Bragg angle. Greater micro strain in ZnO-CuO NCs was found at higher concentrations of mangosteen leaf aqueous extract (from 1.35 × 10^−4^ to 2.77 × 10^−4^) and calcination temperatures (from 1.69 × 10^−4^ to 2.77 × 10^−4^), which was contradictory to when more Cu(NO_3_)_2_·3H_2_O was added (from 2.77 × 10^−4^ to 1.63 × 10^−4^).

### 3.5. FE-SEM Spectroscopy Analysis

The particle size of ZnO-CuO NCs was in the range of 39.10 to 74.53 nm, as tabulated in Table 4. The particle size decreased at elevated mangosteen leaf aqueous extract concentrations (61.46 nm decreased to 39.10 nm) and calcination temperatures (74.53 nm decreased to 39.10 nm). In contrast, a larger particle size was found when 4 g of Cu(NO_3_)_2_·3H_2_O (65.18 nm) was used compared to 2 g of Cu(NO_3_)_2_·3H_2_O (39.10 nm) in green synthesizing ZnO-CuO NCs. The trends of the particle size of the biogenic ZnO-CuO NCs were in accordance with the analyzed XRD results and tabulated in Table 4. The SEM micrographs are shown in Figure 5.

### 3.6. EDX Spectroscopy Analysis

The copper-to-zinc atomic percentage ratio was similar to the copper precursor-to-zinc precursor weight ratio used in synthesizing ZnO-CuO NCs. Neither the mangosteen leaf aqueous extract concentration nor the calcination temperature applied significantly influenced the detected element atomic percentage, as stated in Table 5. On the other hand, compared to 2 g of Cu(NO_3_)_2_·3H_2_O, an obvious increment in copper atomic percentage (18.44%) and its intensity (around 8 keV) were observed by using 4 g of Cu(NO_3_)_2_·3H_2_O in synthesizing ZnO-CuO NCs. The previous copper atomic percentage was only 12.55%. Overall, the synthesized ZnO-CuO NCs depicted the highest atomic percentage in oxygen (60.16–66.25%), followed by zinc atomic percentage (20.10–26.98%) and copper atomic percentage (11.64–18.44%). The presence of an oxygen peak indicated zinc and copper were in oxidized form, and no impurity was found in EDX spectra (Figure 6).

### 3.7. Comparison with Other Studies

The lowest energy bandgap, and smallest crystalline and particle sizes of the mangosteen leaf aqueous extract-mediated ZnO-CuO NCs were selected to compare with other reports, as shown in Table 6. By using less Cu(NO_3_)_2_·3H_2_O, the selected ZnO-CuO NCs’ energy bandgap, crystalline and particle sizes was comparable to other reports. This proved that ZnO-CuO NCs green synthesized in the current study were more cost-effective and eco-friendly when using a mangosteen leaf aqueous extract.

## 4. Discussion

The appearance of an absorption peak at 479 cm^−1^ in mangosteen leaf aqueous extract can be attributed to the *π* → *π** transition [36]. On the other hand, the absorption peaks at 305 cm^−1^ and 308 cm^−1^ indicated the *d*-*d* transition of the Cu(NO_3_)_2_·3H_2_O and Zn(NO_3_)_2_·6H_2_O, respectively. The presence of phytochemicals in the mangosteen leaf aqueous extract led to the occurrence of surface plasmon resonance (SPR) phenomena at a specific wavelength. The change in color of the leaf aqueous extract from light brown to brown upon the addition of the precursors revealed the reduction of zinc(II) ions to zinc(0) and copper(II) to copper(0), followed by oxidation into ZnO-CuO [17,37,38]. As a result, the absorption peaks of the ZnO-CuO NCs were red-shifted to a higher wavelength due to the formation of secondary electronic states, influenced by the metal oxide conjugation with electronic transitions between the valence band and conduction band and the exchange interaction of *s*, *p*-*d* spin within the atoms of metal and oxygen [23].

Regarding the energy bandgap, Ma et al. (2019) [39] reported that the copper precursor did not significantly affect it. However, Fouda et al. observed a significant decrease in the energy bandgap of ZnO-CuO NCs with an increase in the copper precursor amount [40]. Similarly, a decreasing trend in energy bandgap was observed when higher leaf aqueous extract concentrations and calcination temperatures were applied. According to the energy bandgap theory, the energy bandgap of NCs should increase or decrease due to the splitting of each level into a number of levels equal to the number of interacting atoms. In the case of hetero-structured NCs, the bands may overlap [41]. Moreover, the energy bandgap of ZnO-CuO NCs involved coupled transitions from the O_2_ (*2p*) valance band to zinc(II) (3*d*^1^–4*s*) and copper(II) (3*d*^9^) ion conduction bands [42]. Additionally, the presence of CuO, acting as an impurity, reduces the energy bandgap in ZnO-CuO NCs [6,23], and this effect became more significant with higher concentrations of mangosteen leaf aqueous extract and higher calcination temperatures, suggesting the presence of a higher amount of CuO in ZnO-CuO NCs. Also, the redshift in the energy bandgap could be attributed to the interactions between electrons in the localized *d*-orbital of copper ions, which replaced zinc ions and the band electrons in the NCs [43]. This phenomenon makes the NCs efficient in light harvesting for photocatalytic applications [21].

The high PL indicates significant recombination of charge carriers, while low PL suggests maximum charge separation, which is beneficial for the photo-degradation of the processes [6,21,23,42]. The emission peaks of the ZnO-CuO NCs in the violet region (390–405 nm) were attributed to near-band-edge (NBE) emission caused by the defect states in ZnO and CuO [6,21,42]. Furthermore, the lower separation of charge carriers observed in ZnO-CuO NCs calcinated at high temperatures could be attributed to the reduced presence of oxygen vacancies, leading to the enhancement of NBE emission intensity [42].

The phytochemicals present in the mangosteen leaf aqueous extract, such as xanthones, flavonoids and terpene [30,31,32,33], were responsible for the observed functional groups. These compounds played a crucial role as capping, stabilizing and reducing agents during the green synthesis of ZnO-CuO NCs, primarily through electrostatic and steric stabilization mechanisms [32,44]. The vibration of the CuO and ZnO bonds was supported by previous studies [4,5,6,40,45,46]. The bands corresponding to metal oxides and hydroxides are typically located below 1000 cm^−1^ (fingerprint region) due to interatomic vibrations [47]. The sharp band observed in the Zn-O bond vibration confirmed the presence of a strong hexagonal-wurtzite single-phase of ZnO [18]. Additionally, the absence of Cu_2_O could be inferred from the location of Cu-O bond vibration [42], as depicted in Figure 3. Furthermore, slight shifts in the bands indicated structural changes in ZnO-CuO NCs due to the incorporation of an additional element [43]. Changes in the intensity of the bands may be attributed to the variations in the interaction of functional groups from the plant extract under different controlled parameters.

The XRD patterns shown in Figure 4 confirmed the successful biosynthesis of ZnO-CuO NCs [35,40]. Previous literature reports have suggested that NCs with less than 15% of copper exhibited a one-phase wurtzite-like Cu_x_Zn_1−x_O, while those with a higher copper content appeared as a tenorite-like oxide phase, Zn_x_Cu_1−x_O [20,23]. The higher peak intensity of ZnO peaks compared to CuO peaks indicate a higher percentage of ZnO in ZnO-CuO NCs [20,22]. Furthermore, the role of ZnO as a coating material led to lower peak crystallization of CuO [22]. The highest intensity at (1 0 1) corresponded to a ZnO crystal structure grown in the *a*-direction [39]. The intensity of indexed CuO peaks was highest when 4 g of Cu(NO_3_)_2_·3H_2_O was used, indicating the contribution of copper to the formation of ZnO-CuO NCs [21,35,40], which also reflected its higher weight percentage [45,46]. The crystalline size of the ZnO-CuO NCs was similarly reported in Adeyemi et al.’s study [20]. The decrease in crystalline size of the ZnO-CuO NCs with increasing concentrations of mangosteen leaf aqueous extract and calcination temperatures demonstrated the effectiveness of phytochemicals in the plant extract for capping and stabilizing the ZnO-CuO NCs [48], particularly when a high concentration of mangosteen leaf aqueous extract and high calcination temperature were applied. In contrast, a slight increase in the crystalline size of the ZnO-CuO NCs was observed when more Cu(NO_3_)_2_·3H_2_O was added, indicating that the crystallinity of the synthesized NCs was greatly influenced by the variations in the precursor added [18]. However, these results differed from those reported in Fouda et al.’s study, where the crystalline size of their ZnO-CuO NCs decreased with the addition of more copper precursors during synthesis [40]. The broadening of peaks in the XRD pattern of ZnO-CuO NCs was caused by the strain resulting from non-uniform lattice distortion and crystal phase dislocation due to the mismatch in the sizes of zinc and copper atoms [35,42]. Consequently, the presence of a greater number of interfaces in each volume led to a smaller crystalline size [34], and the level of micro strain in the synthesized material increased as the size decreased [42], which was consistent with the results obtained from the calculated crystalline size.

Agglomerated spherical nanostructures were observed in mangosteen leaf aqueous extract-mediated ZnO-CuO, as depicted in Figure 5. This can be attributed to several factors, including the high viscosity of the plant extract [49], the surface physicochemical characteristics [50,51,52,53], the strong forces of attraction between particles [44,54], and the oxidation of metal oxide NPs or NCs [55]. The agglomeration of ZnO-CuO NCs was also influenced by the reduction of salt precursors to zinc and copper ion nucleation mediated by the mangosteen leaf aqueous extract, indicating their role as capping and reducing agents during the formation of ZnO-CuO NCs [43]. The formation of spherical nanostructures (0.05 g/mL) progressively occurred with increasing concentrations of mangosteen leaf aqueous extract, transitioning from irregular nanostructures at low concentrations (0.02 g/mL) of leaf aqueous extract. This may be due to greater isotropic aggregation at the isoelectric point, resulting in strong particle cohesion and the formation of nearly spherical structures [53,56,57] accompanied by the coarsening and coalescence of the NCs [9,11].

Similar results have been reported in terms of EDX analysis by Elemike et al.’s study [23]. The presence of only zinc, copper and oxygen peaks in all ZnO-CuO NCs in the EDX spectra suggests the purity of the green-synthesized ZnO-CuO NCs [20].

Although Yulizar et al.’s study [6] suggested crosslinking between zinc hydroxide and copper hydroxide in the formation of ZnO-CuO NCs, the mechanism and bonding involved in the green synthesis of ZnO-CuO NCs were not clearly addressed by researchers. Phytochemicals present in plants with functional groups, such as -C-O-C-, -C-O-, -C=C- and –C=O- in flavonoids, alkaloids, phenols and anthracenes, have been hypothesized to play a significant role in reducing, capping and stabilizing green-synthesized nanomaterials [58,59]. Xanthones, such as 1, 5, 8-trihydroxy-3-methoxy-2-(3-methylbut-2-enyl) xanthone and 1, 6-dihydroxy-3-methoxy-2-(3-methyl-2-buthenyl)-xanthone, are the major compounds in mangosteen leaf [60]. During chelation, electrons from the precursors’ zinc and copper atoms were donated to form positively charged zinc(II) and copper(II) ions, respectively, which then formed metal complexes with the phytochemicals. These metal complexes subsequently bonded with negatively charged oxygen(II) ions during calcination [61]. Another possible mechanism for the formation of ZnO-CuO NCs was bio reduction, where the divalent oxidation state of zinc and copper were reduced to a zero-valent state by the phytochemicals present in the mangosteen leaf aqueous extract, as indicated by the immediate color change during the green synthesis [62]. ZnO and CuO nuclei were formed after the metallic zinc and copper reacted with the dissolved oxygen in the precursor solution [61], and coordinate covalent bonds were subsequently formed between ZnO and CuO through the lone-pair electron from the oxygen atoms of the metal oxides. A strong framework of ZnO-CuO NCs was then produced during calcination [6]. The possible mechanism and bonding in green synthesizing ZnO-CuO NCs is represented in Figure 1.

## 5. Conclusions

ZnO-CuO NCs were successfully green synthesized using a mangosteen leaf aqueous extract at different concentrations (0.02, 0.03, 0.04 and 0.05 mg/mL), calcination temperatures (300, 400 and 500 °C) and weights of Cu(NO_3_)_2_·3H_2_O (2 and 4 g). The properties of ZnO-CuO NCs were significantly influenced by the green synthesis parameters, including the concentration of the plant extract, calcination temperature and precursor weight. The energy bandgap and crystalline properties of the ZnO-CuO NCs were notably affected by the concentration of the mangosteen leaf aqueous extract and the calcination temperature. However, the intensity of the PL spectrum was solely dependent on the applied calcination temperature. Moreover, the atomic percentage of copper-to-zinc was primarily affected by the weight of the zinc and copper precursor used to synthesize ZnO-CuO NCs. The particle size and morphology were significantly influenced by varied parameters employed in the green synthesis of ZnO-CuO NCs. However, the locations of the FTIR bands in the ZnO-CuO NCs remained consistent throughout the study. The presence of coordinate covalent bonds between ZnO and CuO facilitated by the lone pair of electrons from the oxygen atoms was suggested. The study clearly illustrated the effects of plant extract concentrations, calcination temperatures and precursor amount on the optical, structural and morphological properties during the green synthesis of ZnO-CuO NCs. These findings provide valuable insights for researchers to synthesize ZnO-CuO NCs with specific properties for future applications.

## Data Availability

The data presented in this study are available upon request from the corresponding author.

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
