# Peer review of "Impact of Diverse Parameters on the Physicochemical Characteristics of Green-Synthesized Zinc Oxide–Copper Oxide Nanocomposites Derived from an Aqueous Extract of Garcinia mangostana L. Leaf"

_materials, 2023, doi:10.3390/ma16155421_

Round 1
Reviewer 1 Report
Dear Authors,
1. Fig.5. Are there really FE-SEM images? If this is indeed the case, then they are of very unsatisfactory quality. Small particles are hardly noticeable in the pictures.
2. Why the crystalline sizes obtained with different methods are so different in the Tables 3 and 4?
3. As for figures in the paragraph 3.6. EDX Spectroscopy Analysis and Table 5. Do the authors really believe that the conventional EDX has accuracy hundredths of a percent? As far as I know, the margin of error is several whole percent.
4. Fig.6. There unidenified peaks are clearly visible at 2.1 and 3.4 keV. What about them? Are these phosphorus and tin? Then what kind of "purity" of green-synthesis can we talk about?

Minor typos were found through the manuscript.
Reviewer 2 Report
The article "Optical, Structural and Morphological Properties of Zinc Oxide-Copper Oxide Nanocomposites Green-synthesized from Garcinia mangostana L. Leaf Aqueous Extract with Varied Parameters" is an interesting study on a very relevant topic. In general, the article contains interesting and high-quality material. The obtained nanocomposite material was characterized by a wide range of modern methods of analysis. All the results of the analyzes are discussed and described in sufficient detail. The article is well written and prepared. There are a few comments that do not change the positive impression of the work: The summary of the article needs to be revised and include a conclusion on the work done. It is also desirable to present the results of EDX Spectroscopy in the form of mapping
Reviewer 3 Report
Dear authors,
Thank you very much for your interesting manuscript. The manuscript is introduced a new method for the green synthesis of ZuO-CuO NCs, it is lacking of further experiments for application. Besides, the novelty is lacking too and there are a number of research also trying to synthesize green NCs. Therefore, the manuscript need significantly improve the contents and add more data before considering for publication. The details of major and minor issues are as follow
1. The authors should give some results showing the application of ZnO-CuO NCs materials as mentioned in the Introduction “Consequently, enhancement of the surface area, smaller size and formation of heterojunctions ZnO-CuO NCs strengthens the optical and electronic properties [2,3] which boosts the application in environmental remediation, photo-catalysis, fuel cell, solar cell, antibacterial, UV protection and optoelectronics devices [1,4–6].”
2. There has been much research on the green-synthesized ZnO-CuO method, what is the new point in this study and scientific significance compared to previous publications such as https://doi.org/10.1021/acsomega.1c00310
https://doi.org/10.1021/acsomega.2c02687
https://doi.org/10.1016/j.nanoso.2018.09.003
https://doi.org/10.1021/acsomega.9b02857
https://doi.org/10.1016/j.arabjc.2022.103739
https://doi.org/10.3390/molecules27103206
These novelty points should be strongly emphasized in the abstracts, introduction, and conclusion.
3. The inserted images of Figure 1 should be clearly edited graphs and content.
4. You should carefully check the oscillating signals and properly worded in the FTIR Spectroscopy Analysis part.
5. Figure 2 should be edited, the axis and scale names are not good, too many numbers are detailed, and the different oscillator signals cannot be distinguished.
6. All Figure's resolutions should be improved and provide a higher quality of figures.
7. Full lattice faces should be added in Figure 4.
8. The content details provided in Figure 5 are too much and are not clear, so they should be presented and arranged appropriately.
The authors should intensively check the English since it was tough to read and understand the manuscript in its current form. Therefore, extensive editing of the English language is required.
Round 2
Reviewer 3 Report
Thank you for your revision.
Most of the questions are revised. However, the manuscript is not significantly improved in English. Therefore, the manuscript is better to go through English editing using professional services such as Editage, MPDI English Service, etc.
Besides, Fig 5 is low resolution (out focus). These images must be replaced with a better version.
English is needed to go through the English editing.
